# Child Domestic Work, Violence, and Health Outcomes: A Rapid Systematic Review

**DOI:** 10.3390/ijerph19010427

**Published:** 2021-12-31

**Authors:** Aye Myat Thi, Cathy Zimmerman, Nicola S. Pocock, Clara W. Chan, Meghna Ranganathan

**Affiliations:** 1Innovations for Poverty Action, Yangon 11111, Myanmar; 2Department of Global Health and Development, London School of Hygiene and Tropical Medicine, London WC1E 7HT, UK; cathy.zimmerman@lshtm.ac.uk (C.Z.); nicola.pocock@lshtm.ac.uk (N.S.P.); meghna.Ranganathan@lshtm.ac.uk (M.R.); 3Lumos Foundation, London EC3R 8NB, UK; 4Independent Consultant, London SE1 8UG, UK; clara.chan@phe.gov.uk

**Keywords:** child domestic worker, violence, health, low-income countries, middle-income countries

## Abstract

This rapid systematic review describes violence and health outcomes among child domestic workers (CDWs) taken from 17 studies conducted in low- and middle-income countries. Our analysis estimated the median reported rates of violence in CDWs aged 5–17-year-olds to be 56.2% (emotional; range: 13–92%), 18.9% (physical; range: 1.7–71.4%), and 2.2% (sexual; range: 0–62%). Both boys and girls reported emotional abuse and sexual violence with emotional abuse being the most common. In Ethiopia and India, violence was associated with severe physical injuries and sexual insecurity among a third to half of CDWs. CDWs in India and Togo reported lower levels of psycho-social well-being than controls. In India, physical punishment was correlated with poor psycho-social well-being of CDWs [OR: 3.6; 95% CI: 3.2–4; *p* < 0.0001]. Across the studies, between 7% and 68% of CDWs reported work-related illness and injuries, and one third to half had received no medical treatment. On average, children worked between 9 and 15 h per day with no rest days. Findings highlight that many CDWs are exposed to abuse and other health hazards but that conditions vary substantially by context. Because of the often-hidden nature of child domestic work, future initiatives will need to be specifically designed to reach children in private households. Young workers will also benefit from strategies to change social norms around the value and vulnerability of children in domestic work and the long-term implications of harm during childhood.

## 1. Introduction

Evidence from around the world indicates that exposure to adverse childhood experiences (ACE) hinders children’s development and wellbeing and can often have lifelong effects [1,2]. Adverse childhood experiences (ACE) are associated with a range of illnesses (e.g., heart disease, lung cancer, sexually transmitted infections), mental health symptoms (e.g., depression, anxiety), and social problems (relationship problems, poor job performance, revictimization or perpetrators) in adulthood [1,2]. Children engaged in child labor, including child domestic workers, are particularly vulnerable to different forms of violence, exploitation, and neglect [3].

Child domestic workers are defined as children younger than 18 years who are engaged in domestic work outside the home of their own family for remuneration (whether paid or unpaid), a portion of whom work in hazardous or exploitative situations akin to slavery [4]. Global estimates suggest that that approximately 17.2 million children work as domestic workers, of whom over half (11.2 million) are aged between 5 to 14 years and 67% are girls [5,6]. There remains very limited data on regional estimates of child domestic workers; however, statistics indicate that Asia contains the most (41%) domestic workers [7] and the second-most (60.7 million) child laborers in the world [8]. In many contexts, domestic work is perceived to be safe and beneficial for children who take jobs in employing households to escape poverty, often as a better option than more hazardous income opportunities or to improve their life prospects [9]. Child domestic work is rarely acknowledged as ‘employment’ since, for many employing households, child domestic workers are often ‘relatives’ or ‘fostered’ children, even though they may be treated differently to other family members [10,11]. Hidden behind closed doors in private households, child domestic workers are often denied the protection of national labor laws and legislation [12].

Household responsibilities for child domestic workers often include cleaning, cooking, and caring tasks, and are similar to those of other domestic workers [4]. Such tasks may be considered harmless but can have adverse consequences for children, particularly when they lack the training, experience, and physical and mental capacities to carry out tasks that are not age appropriate. For instance, many domestic tasks can be hazardous for child domestic workers, such as using sharp kitchen utensils, working in monotonous tasks in an awkward position for long hours, assisting with maintenance work from dangerous heights, caring for sick persons, and handling chemicals [4]. Having unspecified or fluid working hours may mean that child domestic workers have to remain available 24 h per day, seven days per week, which can cause sleep deprivation and exhaustion. Chronic fatigue, especially among adolescents, can lead to accidents and cause headaches, and stress- or depression-related syndromes [13]. It is also not uncommon for child domestic workers to be fed leftovers or less or poorer quality food than the family, which can lead to malnutrition, a state that is especially harmful during child growth periods [13]. As children in circumstances of employment, who are unable to assert their rights, they may also be subjected to harsh methods of discipline (corporal punishment, shouting, deprivation of food etc.) for perceived misbehavior or poor performance [12,14,15]. There are numerous accounts from around the world of severe forms of abuse, including extreme physical violence or sexual harassment and the abuse of child domestic workers by males in the household and other males visiting the household [16]. As a result of feelings of powerlessness and low self-confidence, children often feel unable to reject sexual advances or object to exploitation or abuse [13].

Psychological distress, trauma, and subsequent mental health problems are not uncommon among child domestic workers. Young workers often suffer from isolation and the absence of affection and age-appropriate care, alongside being marginalized in the home and experiencing discriminatory treatment by household members [9]. Young people are also generally unable to manage the feelings and emotions that result from circumstances of neglect [13]. These adverse experiences are often compounded by feelings of bereavement due to family separation and loss of affection [2]. Moreover, few working children are permitted to participate in education or access health or social services [9]. The longer-term pathways and health outcomes of child domestic workers have not been studied, but anecdotal accounts suggest that child domestic workers may transition to adult domestic work or marriage arrangements and youth workers may turn to sex work as a less restrictive, more lucrative option than domestic work [17].

While studies have repeatedly suggested the societal costs of child domestic work [18], measuring adverse childhood experiences among child domestic workers has been challenging, in part, due to the invisible nature of their circumstances and also because of the methodological limitations associated with exploring adverse childhood experiences among particularly marginalized youths [2,19,20]. Nonetheless, a growing number of quantitative studies on child domestic workers are offering prevalence estimates and documented health risks and consequences. This study aims to present evidence on the nature of adverse events (specifically violence) and health outcomes among child domestic workers to inform targeted interventions for child domestic workers. This review focuses on low- and middle- income countries (LMIC) and relevant high-income countries (HIC), including Singapore, Taiwan, Macau, Hong Kong, and Brunei, where domestic work is common [7,21]. This review is part of a program of work focusing on child domestic work in LMICs [20]. The primary objective of the review is to describe and synthesize the evidence on violence and health outcomes associated with child domestic work.

## 2. Materials and Methods

### 2.1. Search Strategy

We searched six electronic databases: MEDLINE, EMBASE, Global Health, Econlit, Web of Science, and the International Bibliography of the Social Sciences. We searched for studies published through July 2019, and search terms and concepts were developed by the research team. We also checked the most relevant websites for relevant grey literature: ILO Labourdoc, Freedom Fund research library, Understanding Children’s Work (UCW), Anti-Slavery International, Save the Children, Population Council, UN agency, and the Young Lives website. Letters, commentaries, conference abstracts, books, and book reviews were excluded. We did not track the citations of included studies because of time constraints. The search methodology is stated in the protocol registered as number CRD42019148702 in the PROSPERO database of systematic reviews [22]. Search terms and the timeline for searches can be found in Appendix A.

### 2.2. Inclusion and Exclusion Criteria

Studies were included if they (1) reported any type of physical, sexual, and emotional/psychological violence and/or work-related disease/injuries of child domestic workers (<18 years); (2) described a subgroup analysis or disaggregated data for child domestic workers; (3) were conducted in LMICs and selective HICs, as mentioned above; and (4) were published in English between 1990 and 2019.

As with the primary focus of the review on violence and health outcomes associated with children working as domestic helpers outside their immediate family in their childhood, studies were excluded if they (1) focused on adult domestic workers only (>18 years) and did not report disaggregated data for outcomes of interest among child domestic workers; (2) included only children or young adults (up to 25 years old) performing household chores or care work in their own homes with immediate family; or (3) featured child domestic worker profiles and health literacy or health care utilization, without reference to any of the relevant outcomes. The screening protocol can be found in Appendix A.

### 2.3. Data Extraction and Critical Appraisal

The details of the process of data extraction used are described in another paper published by the study authors [20]. In summary, studies initially identified were uploaded to Rayyan, and duplicates were removed. Two reviewers (CC and NP) screened study titles and abstracts and selected potentially eligible studies for full-text review according to the inclusion criteria. The same reviewers cross-checked each other’s lists of potentially eligible studies and randomly checked excluded studies. Two reviewers independently carried out full-text reviews. CC created the data extraction form and extracted data from 75% of the included studies, while NP did so for the rest. Disagreements were discussed and resolved during data extraction.

For this current review, AT extracted information from each study, and this included the study setting, country, study population, age, study design and measurement tool, sampling method, and summary estimates of violence and health outcomes. In this review, we focus on the prevalence of health and violence outcomes among child domestic workers.

The overall study quality was appraised using the Joanna Briggs critical appraisal tools (CAT) for the relevant study design. We also assessed the quality of the measurement tools used to capture violence and health outcomes using a measurement quality appraisal tool (QAT) developed in a previous study [23]. For the measurement QAT, we extracted data on the method of assessing the outcomes; information on the validity and reliability of measures and any translation of the survey instrument; modifications for cultural sensitivity to questions, and the method of survey administration. Checklists of both appraisal tools are provided in Pocock et al. [20]. For the overall quality appraisal, studies were scored as follows: 0–50% Poor, 51–75% Moderate, 76–100% Good (Appendix B
Table A1). The measurement tool quality was rated as follows: 0–3 “poor”, 4–5 “moderate”, and 6–7 “good” quality. A value of 0 was assigned for studies lacking information on a particular domain [20] (Appendix B
Table A2).

### 2.4. Data Analysis

We employed a narrative synthesis approach, since the objective of our study was to describe violence and health outcomes, not to explore associations between exposures and outcomes [24]. Different types of abuse and violence examined in the included studies were grouped into physical, emotional, and sexual violence when studies did not report individual type of violence separately. Operational definitions of types of violence and study populations are summarized in the Appendix B
Table A3 and Table A4. Violence prevalence was defined as the proportion of child domestic workers who experienced any form of violence (physical, sexual, emotional). When not mentioned specifically for child domestic workers, proportions of those affected were calculated from absolute numbers and relevant information given in studies [10,25,26,27,28]. When types of abuse were reported in the study (e.g., 8.9% slapped/beaten with bare hands, 2.4% beaten with objects), the abuse type with highest percentage was used (‘slapped/beaten with bare hands’ was taken for physical violence) [26]. Prevalence estimates of emotional and sexual violence from 8 studies, and physical violence from 9 studies were extracted, and median values were calculated in excel, as the data were skewed. We interpreted the findings based on recent literature and the quality of study and measurement tools. A meta-analysis was not conducted due to heterogeneity in the study population, definitions, outcomes measured, and methods of assessing health and violence. The prevalence of interested outcomes was calculated from the reported absolute numbers when not specifically described for child domestic workers in the study.

## 3. Results

After the removal of duplicates, we identified 6573 records (see Figure 1). After the study titles and abstracts had been screened, 211 studies were selected for a full-text review. Finally, a total of 17 full studies based on 16 studies and articles were included in the review.

### 3.1. Study Characteristics

Of the seventeen studies included, 10 reported violence and health outcomes, five reported health outcomes only, two reported violence outcomes only, and 12 reported working and living conditions [10,12,14,15,25,26,27,28,29,30,31,32]. The findings are summarized in Table 1 and Table 2 by alphabetical order of the authors’ last names. All studies were conducted in low- or middle-income countries. Eight studies were conducted in South Asia and South East Asia, including in India (*n* = 2) [12,27], Pakistan (*n* = 1) [30], Bangladesh (*n* = 1) [28], Thailand (*n* = 1) [25], Vietnam (*n* = 1) [10], India and the Philippines (*n* = 1) [15], and Cambodia (*n* = 1) [26]. Four studies were conducted in Africa, including in Ethiopia (*n* = 2) [32,33], Senegal (*n* = 1) [34], and South Africa (*n* = 1) [29]; and three were conducted in South America in Brazil (*n* = 3) [31,35,36]. One study was conducted in Haiti (*n* = 1) [37]; and there was one multi-national study conducted in Asia (India, Philippines), Africa (Togo, Tanzania), and America (Peru, Costa Rica) [14]. All studies were cross-sectional surveys with 13 descriptive and four analytical studies [33,35,36,37]. Three out of the 17 studies used nationally representative samples [29,31,37]. The majority of the study populations were child domestic workers (10 studies) [10,12,25,26,27,28,29,30,32,37], while the remaining studies included non-domestic child laborers and children in the community as comparison groups to child domestic workers [14,15,31,33,34,35,36]. Twelve of the 17 studies stated clear definitions of child domestic workers, although their age cut-offs differed. Fifteen studies defined children as being younger than 18 years [10,12,14,15,25,26,28,29,31,32,33,34,35,36,37], two as under 15 years [27,30], and five studies did not provide a specific definition for child domestic workers [31,33,34,35,36]. Study populations definitions are shown in Appendix B
Table A4.

In eight studies, over two-thirds (77–95%) of the child domestic worker study population was represented by girls. One study included adolescent girls only [34], five studies had approximately equal numbers of males and females [26,33,35,36,37], in one study, the majority (62%) of the CDWs were boys (South Africa) [29], and two studies did not provide a gender composition [14,31]. Eight studies [14,15,31,33,34,35,36,37] reported comparisons of outcomes between child domestic workers and non-child domestic workers–non-workers or working children engaged in other sectors, and the remaining nine studies investigated proportions of child domestic workers [10,12,25,26,27,28,29,30,32]. Four studies [15,35,36,37] used odds ratios (OR), adjusted prevalence ratios (APR), adjusted odd ratios (AOR), and prevalence ratios (PR) as statistical methods to assess outcome differences between two groups. Studies were published between 2002 and 2018 [10,12,14,15,25,26,27,28,29,30,31,32,33,34,35,36,37].

### 3.2. Violence Outcomes

Twelve of 17 studies assessed violence prevalence among child domestic workers aged under 18 years [10,12,14,15,25,26,27,28,29,30,32,37]. The median violence rates were 56.2% (emotional; range: 13–92%), 18.9% (physical; range: 1.7–71.4%), and 2.2% (sexual; range: 0–62%). By region, Asia had lower reported violence rates for most types of violence, except for emotional violence, than Africa and America, although the majority of child domestic workers were from Asia (data not shown) and we cannot account for reporting biases.

#### 3.2.1. Emotional Violence

Studies conducted in India, the Philippines, Cambodia, Vietnam, Ethiopia, and Haiti reported emotional or psychological abuse [10,12,15,26,27,28,32,37]. The prevalence ranged from 13% to 92% among former or current child domestic workers [10,12,15,26,27,28,32,37]. In Haiti, male former child domestic workers appear to be at higher risk of being emotionally abused [OR: 3.06; 95% CI: 1.99–4.70; *p* < 0.0001] compared with their female counterparts [OR: 2.41; 95% CI: 1.80–3.23; *p* < 0.0001] [37]. In Ethiopia, physical violence was reported by over 90% of Ethiopian child domestic workers, of whom the overwhelming majority expressed that they were often depressed (57/62) and fearful (58/62) [32]. Child domestic workers in Cambodia (78.2%) were mostly scolded or cursed (91%). Eighty-eight percent (81/92) of those in Ethiopia said that their employer had put a curse on them, which they feared would come true [26,32].

#### 3.2.2. Physical Violence

Nine studies investigated physical violence among child domestic workers [10,14,25,26,27,28,29,30,37]. The prevalence of physical violence among child domestic workers ranged from 1.7% (Thailand) to 71.4% (Haiti) [25,37]. Particularly, female former child domestic workers in Haiti have a greater likelihood of experiencing physical violence than male former child domestic workers: [OR: 2.04; 95% CI: 1.40–2.97; *p* < 0.0001] versus [OR: 1.37; 95% CI: 0.88–2.14; *p* = 0.1661] [37]. An Ethiopian study (N = 100) observed that males (11/16, 69%) are more likely to report experiencing physical violence ‘often’ compared with females (32/84, 38%). However, a higher proportion of female child domestic workers (41%, 19/46) than male child domestic workers (19%, 3/16) reported that they suffered from the detrimental consequences of physical violence such as body swelling/bleeding, bruising, and being seriously hurt and unable to work for several days [32]. In West Bengal, approximately half (46.6%) of the sample of current and former child domestic workers (N = 513, 93% of them were females) reported severe forms of abuse that resulted in injuries, with 25% of them having cuts or bruises as a result of violence [12].

There appear to be regional differences in physical violence. In Togo, 49% of children (*n* = 200) reported being physically punished for mistakes, and in India, 35% (*n* = 500) were physically punished, while in Peru, no child domestic workers (*n* = 199) reported abuse, similar to the Philippines, where 58% (*n* = 200) said they were “just talked to” in response to mistakes [14].

#### 3.2.3. Sexual Violence

The prevalence of sexual violence ranged from 0–62% among former or current child domestic workers [10,12,26,27,28,29,32,37]. In Haiti, childhood experience as a domestic worker was considered to be a risk factor for sexual violence [OR: 1.86; 95% CI: 1.34–2.58; *p* = 0.0001] [37]. Findings from India indicated a large disparity in sexual violence between child domestic workers and control children compared to the difference in Togo. For example, when asked if they know someone who has been sexually abused, 25% of child domestic workers vs. 1.2% of controls in India responded “Yes”, while twice as many child domestic workers as controls knew someone who had been abused in Togo [14]. In Ethiopia, nearly one-third of child domestic workers felt sexually insecure at home, which was related to experiences of sexual abuse [32].

In assessing the experience of sexual violence, several studies have measured different forms of sexual violence, including contact or sexual touching, attempted/forced sex (*n* = 1) [37]; non-contact–teasing and flirting (*n* = 1) [10]; both contact and non-contact (mixed)—forced sex, molestation, forced to watch pornography, obscene language, ogling, flirting, promotion of sex (*n* = 2) [12,32]; and unspecified (reported as sexual abuse/violence/harassment) (*n* = 4) [26,27,28,29]. Three studies that measured sexual contact and mixed forms of sexual violence found high prevalence rates (32.2–62%). Five studies that assessed non-contact and unspecified forms of abuse found relatively low levels of reported prevalence (0–3.4%) [10,26,27,28,29].

### 3.3. Health Outcomes

Sixteen studies described health outcomes categorized into physical health (workplace related conditions and nutritional status), behavioral and mental health, and health care seeking.

#### 3.3.1. Physical Health

In a multi-country study that deployed snowball surveys, child domestic workers self-reported having good or very good health in Tanzania (80%), Philippines (65%), Peru (51%), Togo (46%), and India (36%) [14]. In Brazil, child domestic workers had a 1.2 times higher prevalence of experiencing musculoskeletal pain compared with non-working children [adjusted prevalence rate (aPR): 1.17; 95% CI: 1.05–1.31] [35]. Children who reported working in awkward positions were [aPR:1.15; 95% CI: 1.02–1.30] times more likely to have experienced musculoskeletal pain compared with a non-exposed group [35]. A study in Ho Chi Minh city, Vietnam found that 76% of children reported that their health remained the same after working as a domestic worker, and 17% said their health was better [10].

##### Workplace Illness, and Injury

The percentages of child domestic workers who reported work-related illness varied: 7% (*n* = 115) in Thailand [25], 36% (*n* = 100) in Vietnam [10], 67.9% (*n* = 3841) in Bangladesh [28], and 63% (134/213) in Senegal [34]. Patterns of illness varied among these who fell ill. For example, over 70% of child domestic workers in India and Bangladesh reported gastro-intestinal infections and fever, respectively [27,28]. One-third of child domestic workers in Vietnam reported respiratory problems, and 25% reported back pain (25%) and cuts (11%) [10]. Work-related injury and illness were reported by 4% of child domestic workers aged 10–14 years and 7.6% aged 15 to 17 years [31]. However, injuries were more commonly reported by younger workers [10]. Findings from Cambodia indicated that among those reporting injuries (*n* = 293), one-quarter had been slashed by sharp objects, 10% had slipped in the bathroom, and 6.2% reported electrical shocks [26]. Feelings of exhaustion, insomnia, and fear were reported by one-fifth of youths [26].

In Brazil, child domestic workers had the lowest prevalence of workplace injury/illness compared to child laborers engaged in other forms of potentially hazardous work [31]. For example, the study findings indicate that injuries or illness were reported by 6.57% of those involved in domestic work, 8.2% of those involved in street work, 13.85% of those involved in construction, and 14.87% of those involved in hazardous farming [31]. However, results from South Africa suggest a difference in the risk of injury or illness based on paid versus unpaid work, as the proportion of paid child domestic workers who had been injured (8% of 53,942) was double that in other economic sectors (4% of 3, 243,942), such as unpaid housekeeping and family care, unpaid maintenance and cleaning, begging, farming, and collection of fuel and water [29].

##### Nutritional Status

Stunting seemed to affect a substantial number of child domestic workers in the South Asia studies. Among child domestic workers aged 8–14 years, 55% in India [27] and 90.4% in Pakistan [30] were stunted. Nearly one-third of those in Pakistan had severe stunting [30], and among both groups, 5.7–9.4% were affected by a thin/very thin body mass index (BMI) [27,30]. In Senegal, migrant child domestic workers tended to reside in more socio-economically affluent environments, and they had more advanced breast development (*p* = 0.045) and occurrence of menarche (*p* = 0.014) and better nutritional status: higher mid-arm circumference (*p* < 0.001), body mass index (BMI) (*p* < 0.001), and fat mass index (FMI) (*p* < 0.0001) compared with non-migrants in rural areas [34].

#### 3.3.2. Behavioral and Mental Health

Findings from Brazil suggest that child domestic workers in low-income urban areas have a 1.6 times higher prevalence of behavioral problems [aPR: 1.6; 95% CI: 1.0–2.7; *p* = 0.052] than children who do not work [36]. In a multi-national study, the difference in psychosocial wellbeing between child domestic workers and controls in India and Togo was substantial while the disparity was not obvious in the Philippines, Peru, and Tanzania [14]. The proportion of child domestic workers (67%) with psychosocial scores in the lowest stratum was more than twofold that of controls (25%) (*p* < 0.001) in India, while in the Philippines, there was a 6% difference between the two groups (*p* = 0.2) [15]. Importantly, the findings suggest an influence of abuse on psychosocial well-being, as children within the lowest tercile for psychosocial well-being were more likely to have been harshly punished (beaten/deprived of food) in India [OR: 3.6; 95% CI: 3.2–4; *p* < 0.0001] [15]. In a study on nine provinces in South Africa, twice as many child domestic workers feared being hurt by someone (13%) compared with children working in other sectors (7%) [29]. Limitations to socializing were reported by a study in Ethiopia, with 2.5% of child domestic workers (*n* = 100) reporting difficulty in socializing with others compared with 0.3% of non-laborers (*n* = 400) (*p* = 0.006) [33].

#### 3.3.3. Health Care Seeking

In Vietnam and Thailand, approximately half of sick or injured child domestic workers said they were not treated for their injuries [10,25]. Almost all child domestic workers who reported being ill in Bangladesh had received some form of treatment (self-treatment, doctor, pharmacy, traditional healer, treated by employers), even though one-third of them had to work during sickness [28].

### 3.4. Working Conditions

The average number of working hours reported by child domestic workers ranged from 9 to 15 h per day in Bangladesh, Vietnam, India, and Ethiopia [10,12,28,32,34]. Working hours were reported differently across studies, and findings showed that 95% of workers in Pakistan worked overtime (unspecified hours) [30]. High numbers of children worked between ten and twelve hours per day, including 95% in India, 65% in Tanzania, and 52% in Togo [14]. In Thailand, 78% worked more than eight hours per day [25], and in Cambodia, 10.2% said they worked between nine and thirteen hours per day [26]. In Brazil 70% of youths aged between 10 and 17 years old worked more than 30 h per week [31]. Findings from India, Thailand, Cambodia, Vietnam, and Ethiopia indicated that children had no rest days, ranging from 31% in West Bengal, India to 94% in Ho Chi Minh, Vietnam [10,12,14,25,26,32].

In addition to domestic chores, child domestic workers were involved in caring tasks—child and elderly care, washing legs and feet, helping family members to bathe; outdoor chores—gardening, pet feeding, fetching water and fuel, taking children to school, going to market; and employers’ businesses—helping with family businesses and garages, helping to sell commodities in open market petty trade, farming-herding cattle, milking cows, cattle raising, and paltry nursing [12,14,25,26,27,28,30,32]. In Ethiopia, 45% of CDWs carried/lifted heavy loads beyond their capacity, while some handled hot water, hot iron, and sharp knives for chores [32]. Forty-two percent of CDWs in Brazil used machines/chemicals at work, whereas only one-third of them wore protective gear or received training [31]. A total of 14–23% of CDWs in Brazil and South Africa performed heavy physical work, monotonous/repetitive work, and or work in an awkward posture [29,35].

### 3.5. Critical Appraisal of Study Quality and Measurement Tools

Based on the Joanna Briggs critical appraisal tools (Appendix B
Table A1), four studies were scored as “good” [33,35,36,37], three were “moderate” [26,28,34], and ten were “poor” [10,12,14,15,25,27,29,30,31,32]. Studies were rated ‘poor’ mainly because of unclear self-reporting methods, particularly regarding sample size, data analysis, outcomes, and response rates. Likewise, elsewhere [20], overall study quality appraisal scores were different from the critical appraisal ratings of the measurement tools in the majority of studies (Appendix B
Table A2). For four studies that were ranked “good” in terms of the overall study quality, their measurement tools were appraised as poor [33,37] and moderate [35,36]. No studies scored ‘good’ for their measurement tools. Three of the twelve violence studies and four of the fifteen health studies were rated ‘moderate’ for the quality of their measurement tools, while the rest scored ‘poor’ (Appendix B
Table A2).

No studies used an internationally validated screening tool to assess child violence exposures. Most often, authors conceived their own violence questions, or questions were loosely based on the limited questions for violence available in the ILO Statistical Information and Monitoring Programme on Child Labor (SIMPOC) questionnaires, which have not been formally validated. However, three studies [33,35,36] used internationally validated tools to measure health outcomes; two were validated for use with children including the Child Behavior Checklist (CBCL) for behavioral problems [36], the Reporting Questionnaire for Children (RQC) to screen behavioral/mental problems, and the Diagnostic Interview for Children and Adolescents (DICA) to confirm diagnosis for screening positive cases [33]. The Standardized Nordic questionnaire for musculoskeletal symptoms was not originally developed for use with children; however, it has been used in child labor studies [35,38,39]. Health outcome measures, including psychosocial health, were usually developed by the researcher. Limited studies examining occupational health outcomes used questions from ILO SIMPOC model questionnaires, which have not been formally validated.

Ethical approval was not mentioned in five violence and health studies [26,27,28,29,32] or two health studies [31,36]. Only one study stated that it adhered to the WHO guidelines for ethics and safety recommended for research on violence against women [37]. A quarter of studies [14,32,33,37] noted the use of methods to ensure cultural appropriateness.

## 4. Discussion

This rapid systematic review provides a narrative synthesis of the violence, health outcomes, and working conditions of child domestic workers. Child domestic workers are generally excluded from mainstream child protection and education services and are vulnerable to different forms of violence and maltreatment in employing households [9]. Importantly, working conditions and children’s experiences, including exposure to violence, and the circumstances in which children perform different tasks are critical contributing factors to the ways in which child domestic work affects children’s health, development, and safety.

Ultimately, we identified 17 studies conducted in low- and middle-income countries that described violence and health outcomes experienced by child domestic workers. Half of the studies were conducted in Asia, while the rest were conducted in Africa and America. Our analysis estimated that the median reported rates of violence in child domestic workers aged 5–17-year-olds are 56.2% (emotional violence), 19% (physical violence), and 2.2% (sexual violence). By region, Asia had lower median prevalence rates of physical and sexual violence compared with the Americas and Africa. It is, however, difficult to generalize regional prevalence estimates, as only one-third of the 17 studies [10,26,28,29,37] were nationally or regionally representative samples, whereas the remaining studies used convenience or purposive samples. Definitional variations in measuring abuse and violence across the studies also make comparisons difficult.

Across the studies, emotional violence had the highest prevalence (over 50%) compared with other forms of violence, and this also varied by region—over 50% in Asia and North and South America and 92% in Africa. This aligns with estimates by the World Health Organization (WHO) that psychological abuse is the most commonly reported form of maltreatment in a child’s lifetime [40]. This number also echoes the findings from another global systematic review that estimated that over 50% of children (2–17 years) across the world have experienced some form of abuse in the past year [1,41]. Despite a lack of clarity on whether the reported violence occurred over the lifetime or in the past year in many of our included studies, reported prevalence rates indicated that different forms of violence were experienced in employing households. According to the WHO, restricting a child’s movements is considered a form of emotional or psychological violence [42]. Only a few studies in this review documented movement control. For example, findings from Thailand and Ethiopia indicated that a large number of CDWs are restricted to their employers’ premises [25,32]. Accounting for movement restriction in future research may help to estimate emotional abuse and its effects. There can be little doubt that during a child’s development, the absence of caregiving, including emotional support, compounded by emotionally abusive treatment by the main adults in a child’s life, will cause long-lasting damage to a child’s healthy psychological growth and well-being, including feelings of self-confidence.

Verbal abuse is treated as a form of emotional violence if it is continuous and severe and negatively affects an individual’s emotional state [43,44]. Perceived verbal abuse that damages brain development is associated with diverse personality and behavioral disorders and produces long-lasting consequences [43,44]. The effects of emotional violence, either acting alone or together with physical and sexual violence, may be intensified when they interact with pre-existing adverse childhood experiences, such as restricted freedoms, long-term separation, and parental loss. Our review highlights the importance of emotional violence among child domestic workers that is harmful to children but may not be considered as important as physical and sexual violence [44]. During a child’s years of social development, verbal abuse such as repeated insults, criticisms or threats, is likely to have effects that last into adulthood.

Notably, the median prevalence rate of sexual violence among child domestic workers (derived from LMIC studies) was comparatively lower than the WHO’s global lifetime sexual violence prevalence figures of 8% for boys and 18% for girls [40]. In our review, studies that reported sexual contact and mixed forms of sexual violence were more prevalent than those that measured non-contact and unspecified forms, including sexual harassment and abuse. The prevalence may have been affected by the type of sexual violence that was measured, as respondents might have been more likely to recall more serious episodes that involved contact versus those that seemed less harmful or without contact. This finding is similar to results from studies in Ethiopia, in which approximately one-third of young Ethiopian domestic workers below the age of 25 had experienced coerced/forced sex, suggesting that domestic work is a risk factor for non-consensual sex and early sexual initiation [45,46]. However, there remains debate about whether the form of sexual violence affects the reported prevalence [46,47]. Geographical variations and under-reporting due to shame may also affect differences in reported prevalence between studies. Given the likelihood of underreporting and the severe damage to youth who suffer sexual abuse or harassment, there can be no doubt that these types of abuse call for more sensitive forms of investigation and stronger prevention initiatives.

From our review, we cannot draw conclusions on whether violence prevalence rates among child domestic workers differ by sex, as most studies did not collect sex-disaggregated data, because females generally dominate the domestic work sector [48] and are at a higher risk of experiencing sexual violence than males [37,40,47,49]. In this review, the nationally representative Haitian household survey demonstrated that female former child domestic workers have a greater risk of experiencing physical violence, and former male child domestic workers have greater odds of experiencing emotional violence and a similar risk of experiencing sexual violence compared with females [37]. However, the survey measurement tool used in the Haitian study [37] was appraised as poor due to a lack of information on its validity and reliability. Thus, further research is required to support this finding.

The findings of this review add to the evidence that violence against children has consequences for health and wellbeing by specifying the abuse experienced by children in domestic work [50,51]. In our review, physical violence and punishment were shown to cause severe physical injuries (Ethiopia and West Bengal, India) [12,32] and can also be attributed to poor psycho-social wellbeing (India and Togo) [14] and poor self-reported health (India and the Philippines) [15]. For instance, in Ethiopia and Cambodia, the majority of child domestic workers who had experienced violence suffered from depression, fear, insecurity, suspicion, worthlessness, anger, apathy, and insomnia. [26,32]. This aligns with research from a nationally representative study from the United States where children who were physically punished or abused (with or without physical punishment), had increased odds of having two or more psychiatric disorders between the ages of 15 and 54 years old [52].

Our review highlights the critical issue of child domestic workers engaged in hazardous work and working conditions. The child domestic workers surveyed in many studies from Asia and Africa worked more than nine hours per day with no rest days, and those who worked long hours with fewer breaks had poorer psycho-social well-being and a higher incidence of injuries in India and Brazil [15,31]. Research shows that working over 60 h per week increases the risk of mental health problems and cardiovascular diseases [4,53]. The ILO recommends uninterrupted rest daily and a minimum of 24 h rest after working consecutively all week for domestic workers [54]. Although specific evidence and recommendations for children are lacking, age-specific work hours and regular rests are particularly important for children because, biologically, they need longer sleep hours and adequate rest and are prone to fatigue [4,18].

In addition, child domestic workers from studies in this review engaged in physically harmful work (e.g., carrying heavy loads, using machines and chemicals, being exposed to noise, unnatural movements) and mentally exhausting tasks (e.g., caring for children and the elderly), generally without adequate safety measures. Compared with adults, the developing bodies of children are exceptionally susceptible to occupational hazards. For instance, children’s thinner skin easily absorbs high doses of toxics and heavy metals, their rapidly growing skeletons are more vulnerable to unnatural posture and movements, and their premature thermoregulation is more sensitive to temperature. All of these factors predispose them to increased risks of neurobiological problems, immune impairment, non-communicable diseases, musculoskeletal disorders, respiratory problems, and cancer [18,35,55]. In this review, a study from Brazil showed that children working in awkward positions had a higher prevalence of musculoskeletal problems [35].

This review confirms that accidents due to poor working conditions are common. Child domestic workers reported injuries from cuts [10], slashes, electrical shocks, falling from stairs, and sore fingers and toes from detergent use [26]. They may be particularly prone to accidents because of their inability to correctly assess dangers and threats [4] and due to mental and physical exhaustion resulting from overwork, occupational stressors, and violence [18]. Evidence indicates that night work, heavy work, and exposure to physical hazards increase the likelihood of workplace injury in working children by 40% [18]. Subsequently, youth workers have higher rates of occupational injury, illness, and fatality compared with adult workers [18]. Our review also found that child domestic workers suffer from malnutrition, gastrointestinal infection, anemia, stunting [27,30], vitamin deficiencies, skin disease, musculoskeletal problems [10,35], and respiratory problems [10,27]. Child domestic workers also have poor access to care and may not receive the required medical treatment or rest unless the employers permit this. Results indicate that one-third of child domestic workers that reported feeling sick in Bangladesh had to work [28], while approximately half of sick or injured child domestic workers in Thailand and Vietnam did not receive adequate treatment [10,25].

The quality of studies included in our systematic review was variable, as fewer than half of cross-sectional surveys (7/17) were assessed as having medium to good quality study design; however, the measurement tools used to assess health and violence outcomes in studies were scored as moderate to poor. No studies used an internationally validated screening tool to assess child violence outcomes. Most of the measurement tools were conceived by researchers, and study authors provided very limited information on the development of measurement tools. The study quality would have been improved if study instruments drew on validated health and violence measures in LMIC contexts, followed by pilot testing, cognitive interviews, and options for adaptation in different countries. No studies reported cognitive interviews and the majority of studies did not mention the culturally sensitive modification of questionnaires (*n* = 14) or pilot testing (*n* = 12), which are criteria for assessing measurement tools.

Research on violence against children requires particular attention to ethical, safety, cultural, and legal concerns. Questioning young people about abusive experiences may cause youths to recall traumatic experiences, which means that adequate referral mechanisms must be in place to provide the necessary support. Moreover, in many locations, there are also legal child protection reporting requirements. Furthermore, having strong protocols in place to ensure anonymity and confidentiality is essential for the safety of participants. However, seven studies reported no information on ethical or safeguarding procedures for the research, including five violence studies [26,27,28,29,32] and two health studies [31,36]. However, ethics was not applied as appraisal criteria.

## 5. Strengths and Limitations of the Study

To the best of our knowledge, this is the first systematic review to document violence and health outcomes among child domestic workers. However, this review has certain limitations. First, we extracted available heterogenous information of child abuse and violence from the studies that used different sampling strategies to calculate the median violence estimates. We used median estimates as the data were skewed, and mean estimates may not provide accurate estimates. The content and clarity of the questions used to assess violence in the studies differed. Eight studies asked CDWs about the specific types of abuse they had experienced (‘have you ever been punched, kicked, whipped, or beaten with an object, choked, smothered’) [37], while the remaining six reported proportions of child domestic workers who had been ‘physically punished’ or who had experienced ‘mental assault’ or ‘sexual violence’ without specifying the acts included under these categories [14,27]. As with many studies using self-reporting measures, for consistency and accuracy, we excluded violence rates reported through indirect questions (‘know someone who has been abused’) [10,14]. We also recognize that children may be scared to report honestly about abuse, especially sexual abuse, due to fear or shame. For these reasons, the median violence estimates from this review are likely to be underestimates.

Second, the impact of violence on health and well-being escalates with the degree of adverse experiences for children exposed to abuse. Exposure to one ACE doubles the risk of poor health, and experiencing more than four ACEs triples the risk of poor health compared to children with no exposure at all [51]. Furthermore, the consequences of inter-related-abuse may cause stunting, which in turn predisposes individuals to low self-esteem and other behavioral problems [56]. Given the likelihood of multiple interacting proximal and distal factors being associated with violence and maltreatment, it is difficult to disentangle the effects of violence associated with child domestic work. For example, child domestic workers may be affected by other adverse experiences such as separation or loss of parents, chronic poverty, and domestic violence in their birth family, in addition to violence at their employing household.

Third, there is no clear consensus on the difference between physical punishment and physical abuse [57]. Physical punishment, in many contexts, is considered a normal disciplinary tool, while abuse is considered to be harmful for child health and development [57]. However, these forms can often overlap, because physical punishment as discipline may also be harsh and harmful [58]. This review shows that one-third to half of child domestic workers surveyed in Ethiopia and India (West Bengal) had experienced bodily injuries due to what was recorded as physical punishment. As it is difficult to distinguish between physical punishment and abuse in these studies, beating, hitting, and slapping were considered physical violence, even though these behaviors may be committed for correctional purposes, which would have influenced the reported rates of physical violence.

Fourth, this review did not observe any significant association between behavior and mental health problems and child domestic work, although both conditions were more common in child laborers in the two studies [33,36]. This may be because both studies were unable to adjust for the effect of violence and child abuse on mental health problems among child domestic workers [33,36]. If child domestic workers have been abused, the impact of this on health may appear in early childhood or later in adult life [36,51]. For instance, a study in New Zealand which prospectively followed up children exposed to physical and sexual abuse in childhood, found associations with the mental disorders depression, anxiety disorder, conduct disorder, substance use, and suicidal tendency, which appeared between the ages of 16 and 25 years [59]. As the studies in our review are cross-sectional studies, this kind of longitudinal causal associations could have been missed. Finally, similar to the paper by Pocock et al., we assessed the study quality and measurement tools based on the authors’ reporting, and we were unable to distinguish whether poor scoring was due to incomplete reporting or poor study design or measurement tools [20].

## 6. Implications for Research and Programming

Our findings demonstrate that child domestic workers are more likely to be exposed to various forms of violence and occupational hazards compared with child workers in other sectors or non-working children. Importantly, violence exposure appears to influence whether domestic work increases the risk of adverse health outcomes in children. Despite extrapolating the idea that occupational risks are harmful for children from the data available, we cannot provide conclusive evidence on which elements of work and frequency and severity of hazardous work can threaten the health, safety, and well-being of child domestic workers. Simply asking tasks of child domestic workers is not enough, because studies need to ask specifically about what, where, how, and how long they work on each task for to determine occupational hazards. Because of the wide range of work-related risks, e.g., harsh chemicals, sharp knives, cooking, especially for small, growing bodies, future research should explore occupational hazards relevant to child domestic work. Longitudinal studies that follow child domestic workers into adulthood may be required to determine and differentiate the effects of child domestic work and abuse on health and provide well-informed child protection strategies [60].

Our results also indicate the need for programming that is specifically designed to reach children who are involved in domestic work. Because these youths work in relative isolation, away from public view, initiatives to address emotional, physical, or sexual abuse and promote healthy child development will have to address the behaviors of employers while identifying the most effective ways to provide support to abused youths. Community-based violence-prevalence interventions that include poverty reduction alongside psycho-social activities in areas where there is a high density of CDWs may have promise [61]. However, these strategies may also exclude youth who are the least visible and hardest to reach. Currently working youth may benefit from combined interventions that aim to change social norms around child domestic work and simultaneously provide young workers with useful skills training and viable opportunities to improve their future livelihoods.

## 7. Conclusions

In conclusion, our review highlights the associations between child domestic work, violence, particularly emotional violence, and effects on health. Our results also suggest the poor working conditions and occupational hazards that place these young workers at risk of accidents and injuries. Ultimately, our findings suggest the need for greater attention and more strategic action to protect young people in situations that are often hidden from view. Behavioral change interventions that identify and shift the harmful norms and behaviors and increase awareness about children’s vulnerability to occupation hazards targeted to employing households may improve the living and working conditions of young workers through changing the social acceptance and tolerance of violence and exploitation of child domestic workers.

## Figures and Tables

**Figure 1 ijerph-19-00427-f001:**
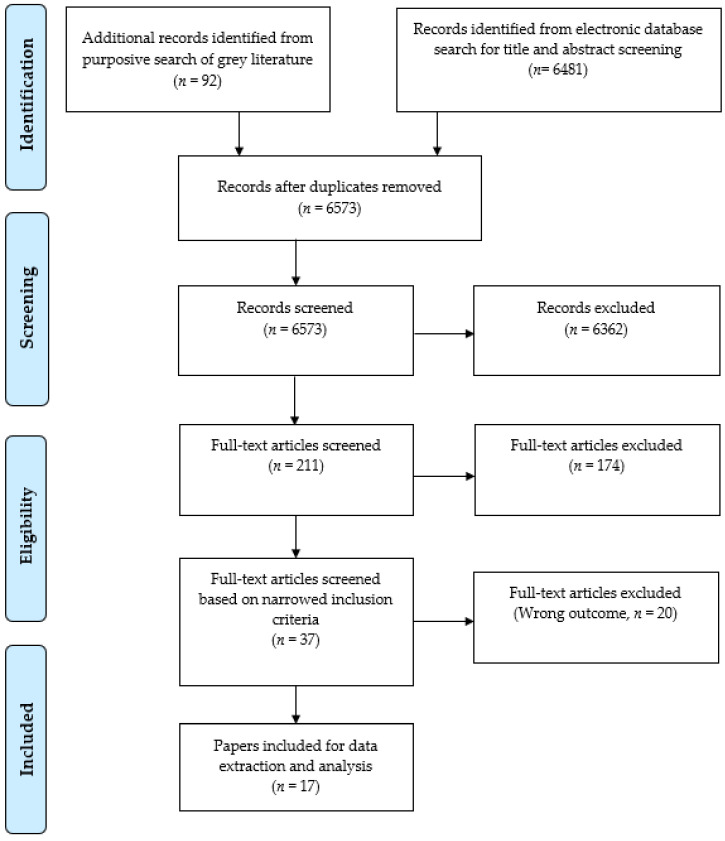
PRISMA flowchart of the CDW violence and health study selection.

**Table 1 ijerph-19-00427-t001:** Overview and characteristics of studies that reported violence and/or health outcomes among child domestic workers (CDW) (*n* = 17).

No	Study	Country	Setting	Study Population/Sample Size	Age (Years)	Study Design and Measurement Tool	Sampling Method	Primary Outcomes (Violence and Health Outcomes)
1	ACPR, ILO 2006	Bangladesh	Urban and rural areas of Bangladesh including five cities (Barisal, Chittagong, Khulna, Rajshahi, Sylhet) excluding Dhaka and the remaining urban and rural areas of Bangladesh	CDWs (*n* = 3, 841) in employer households (*n* = 3, 805) surveyed from December 2005 to February 2006 (estimated number of CDWs in Bangladesh:273,543)	5–17	Cross-sectional population-based household survey with CDWs and employers in selected households from December 2005 to February 2006	725 urban and rural primary sampling units (PSU) were selected from 5 cities (excluding Dhaka) using the circular systematic method with probabilities proportional to size. Segments of PSU were purposively selected from CDW-concentrated areas and randomly selected from the remaining areas. CDW households were selected from high CDW concentration areas (6 households) and from other segments (4 households) via simple random sampling without replacement.	Abuse (scolding, slapping and beating), sexual violence, work-related illness and treatment seeking
2	Alem 2006	Ethiopia	Four industrialized cities in Ethiopia: Addis Ababa and three other regional towns—Nazareth, Awassa and Bahirdar.	Children (*n* = 2400): Child laborers (*n* = 2000 including 100 CDWs) and neighborhood controls (*n* = 400) (no exact data for findings though there are aggregated findings)	8–15	Cross-sectional household survey consisting of two structured questionnaires applied in a two-stage design from October 2001 to May 2002	Systematic (probability) sampling of households was applied to recruit child laborers and neighborhood controls. Based on 1994 census data, initial household selected using random numbers method, and subsequent households were visited to recruit domestic workers until a sample of 20 were reached for each of the five selected study sites (kebele). Purposive sampling was used to recruit street and commercial sex workers.	Mental, behavioral, physical, and nutritional problems amongst child laborers, versus non-laborers.
3	Banerjee 2008	India	Kolkata, India	CDWs (*n* = 330)	8–14	Household survey. (timeline not reported)	2500 households surveyed—no information on sampling method.	Physical, emotional and sexual abuse, disease and nutritional status amongst CDWs
4	Benvegnu 2005	Brazil	Low-income areas of Pelotas, Southern Brazil	Children and adolescents (*n* = 3139) from low- income households: child workers (*n* = 434) including 89 young domestic workers	10–17	Standardized pre-coded questionnaire, cross-sectional household survey. (2002)	Random selection of low-income areas in Pelotas (22 of 70 neighborhoods), based on census data—all children (10 to 17) living in households in these areas were interviewed (excluding homeless or street children).	Prevalence of behavioral problems amongst child laborers compared to non-workers
5	Budlender and Bosch 2002	South Africa	Nine provinces in South Africa	Child laborers (*n* = 3,476,358): CDWs (*n* = 53,942)	5–17	Cross-sectional household survey (SIMPOC SAYP), from June to July 1999	Household surveys in 30,550 households across 9 provinces, which provided information on 33,000 children aged 5–17, (first phase.) Second phase: probability sub-sampling for detailed survey on activities of children from 6110 households containing at least one child doing work of some kind, collected information on approximately 10,000 children. A second source of data was a Time Use survey (2000) based on the sample frame of the SIMPOC SAYP survey, with over 8500 households sampled. Results for both phases were weighted to make them representative of the entire population of 5–17-year-olds.	Sexual violence, work-related injuries, illness and complaints
6	Degraff 2016	Brazil	Brazil	Children (*n* = 60678): CDWs (*n* = 1129) out of children engaged in hazardous forms of labor (*n* = 2608) 16.7% of children engaged in child labor; 25.7% of these in risky work; 43.2 of these in domestic work	10–17	Nationally representative household survey (PNAD 2001)	Secondary data analysis of Brazil’s 2001 annual household survey, the Pesquisa Nacional por (PNAD 2001). The PNAD-2001 is a nationally representative sample survey including 126,898 households and 378,837 individuals. Of this population, children aged 10–17 years and their families were focused.	Occurrence of work-related injury/illness
7	Fassa 2005	Brazil	Low-income areas of Pelotas, Brazil	Children and adolescents from low-income households (*n* = 3269): Child workers (*n* = 451) including 105 CDWs (no crude figure reported, calculating from percentage—24.1%—105/404 of child workers are CDWs)	10–17	Cross-sectional household survey conducted from January to June 1998.	Random selection of low-income areas in Pelotas (22 of 70 neighborhoods), based on census data—all children (10 to 17) living in households in these areas were interviewed (excluding homeless or street children).	Prevalence of musculoskeletal pain at various anatomical sites amongst different categories of child laborers in the preceding 12 months
8	Gamlin 2015	Peru, Togo, India, Tanzania, the Philippines, Costa Rica	Selected districts in six LMICs (Peru, Togo, India, Tanzania, the Philippines, Costa Rica)	Children (*n* = 3062): CDWs (*n* = 1465) and neighborhood controls (*n* = 1597)	6–18	100-item questionnaire (developed by research team in collaboration with ASI—based on findings from systematic review and qualitative study), administered by local research teams and partner organizations in selected districts from April to October 2009	Opportunistic sampling-participants recruited through NGOs, schools and the neighborhood snowball technique.	Physical and sexual abuse and psycho-social wellbeing of CDWs
9	Garnier 2003	Senegal	Niakhar (rural district) and four urban cities (Dakar, Mbour, Fatick and Joal), Senegal	Adolescent girls (*n* = 331) who migrate to the cities to work as maids (*n* = 213) or stay behind in the rural villages (*n* = 119)	14.5–16.5	Cross-sectional household survey as a part of the Cohort study “growth during adolescence” conducted from April to June 1999	Study participants were from 30 selected villages in Naikhar district, identified through the framework of a longitudinal study of all adolescent girls born and raised there up to the age of 10 (from 1995 to the time of study). No information on how villages were selected. Unclear if participants are representative of the total population as no attrition rate reported (possibility of selection bias).	Morbidity and healthcare behaviors during the 3 months prior to the survey. Sexual maturity, nutritional status, and health of migrant maids versus non-migrants staying in rural villages
10	Gilbert 2018	Haiti	Haitian households displaced by 2010 earthquake (including IDP camps)	CDW (ever been restaveks—child domestic servants who perform unpaid work) (*n* = 451)	13–24	Nationally representative cross-sectional household survey of children and young people (Violence Against Children Survey 2012) conducted from May to June 2012	Stratified, three-stage cluster design used to sample households and camps affected by the 2010 earthquake. Sample based on updated estimates from the 2003 Haitian census	Reported violence before 18 years (physical, emotional, sexual) amongst CDW vs. non-CDWs
11	Hesketh 2012	India, the Philippines	Selected states/cities in India (Tamil Nadu, Kerala, Maharashtra, Andhra Pradesh, Uttar Pradesh and Bihar) and the Philippines (Manila, Batangas, Bocolod, Cebu, Davao, Dumaguete and Iloilo)	Children (*n* = 1400): CDWs (*n* = 700: 200 Flipinos and 500 Indians) and school-attending neighborhood controls (*n* = 700: 200 Flipinos and 500 Indians)	<18	Cross-sectional survey—20 item questionnaire (developed by research team in collaboration with ASI—based on findings from systematic review and qualitative study), administered by local research teams and partner organizations in selected districts over a six-month period (for control group, mostly self-administered) from April to October 2009	Opportunistic: participants recruited through NGOs, schools, and neighborhood snowball technique.	Physical abuse and psycho-social wellbeing of CDWs
12	ILO 2006	Vietnam	Ho Chi Minh, Vietnam	CDWs (*n* = 100), employers (*n* = 10) and parents (*n* = 8)	6–17	Structured surveys with CDW, employers and guardians from April to November 2005	The sampling frame was lists of 100 clusters of households randomly selected from a total of 8989 clusters of households from the 8 selected (out of 24) districts. CDWs were identified from these lists. As the random sampling found only 20 CDWs from the 100 clusters of households, another 100 clusters of households were randomly selected from the pool of clusters of households following the same sampling procedure.	Reported violence, self-assessed physical health and injury
13	Kifle 2002	Ethiopia	3 districts (woredas) in Addis Ababa, Ethiopia	CDWs (*n* = 100)	<18	Quantitative survey in the form of structured questionnaires and qualitative methods (focus group discussions with key informants, in depth interviews, role plays). (2002)	Rapid assessment methodology and purposive sampling. Potential participants were recruited by facilitators/enumerators familiar with the study sites from areas frequented by child domestic workers, including schools, market places, literacy centers, water points, and domestic employment broker stands.	Physical, emotional violence (verbal abuse) and sexual harassment and medical treatment pattern
14	NIS Cambodia, ILO 2004	Cambodia	slum and non-slum areas in the seven districts of Phnom Penh, Cambodia	Live in CDWs (*n* = 293)	7–17	Household survey conducted from September to October 2003.	Simple random sampling method used in 125 villages (primary sampling units—PSU) selected Fixed sample sizes of 20 households (secondary sampling units) were chosen from each PSU using linear systematic sampling with a random start. Selected households (*n* = 2500) were surveyed to identify the presence of CDWs and detailed interviews were conducted with adult members,	Type of punishment, sexual violence, injuries among CDWs
15	Phlainoi 2002	Thailand	Children from north-eastern, central, northern, and southern regions in Bangkok, Thailand	CDWs (*n* = 115)	12–17	Cross-sectional survey based on Rapid Assessment Methodology from ILO/UNICEF. (2001)	Purposive sampling. Surveys with CDWs in Bangkok (24/50 districts) and in original villages in Northeast Thailand. CDW employer households were identified via Bangkok Metropolitan Council Members who had access to communities and teachers who asked students whether there were CDWs in their households or communities.	Physical violence, work associated diseases and injuries among CDWs
16	Save the Children UK 2006	West Bengal	Four districts across West Bengal (districts not reported)	Current or former CDWs (*n* = 513)	<18	Quantitative survey in the form of structured questionnaires and qualitative semi-structured interviews conducted from April to September 2005	Of 1020 former/current CDWs who participated in the Save the Children UK’s projects, those who had been engaged with the organization’s “drop-in non-formal education centers” for at least three months were recruited.	Reported physical/sexual/emotional abuse and violence
17	Zainab and Kadir 2016	Pakistan	Squatter settlements in Gulshan town of Karachi, Pakistan	Live-out CDWs (*n* = 385)	10–14	Cross-sectional household survey from May to October 2011	Random selection of 6/46 squatter settlements. Recruitment through non-probability snowball sampling technique	Physical abuse at their workplace in the past six months, and nutritional status among CDWs

ACPR—Associates for Community and Population Research, ILO—International Labour Organization, CDW—child domestic worker, SIMPOC—Statistical Information and Monitoring Programme on Child Labour, SAYP—Survey of Activities of Young People, PNAD—Pesquisa Nacional por Amostra de Domicílios, LMIC—low and middle income countries, ASI—Anti-Slavery International, NIS—National Institute of Statistics.

**Table 2 ijerph-19-00427-t002:** Summary description of main findings from the studies that reported violence and/or health outcomes among child domestic workers (*n* = 17).

No	Study	Summary Estimates	Main Findings
1	ACPR, ILO 2006	Of those 3841 CDWs, 60.3% had experienced abuse; 39.7% reported no abuse 59% reported good treatment by their employer	Violence: Of those who were abused: 60.1% were scolded, 19% were slapped or beaten, 0.6% reported sexual violence. Health and care during illness 68% had fallen sick at employers’ houses. Among those who had ever fallen ill, 76% had experienced fever, 41.7% cough and cold, 9.7% headaches, 7.3% water-borne diseases Nine out of ten sick CDWs had received some kind of treatment. Of those who received medical treatment, 31.8% saw a doctor, 66% a pharmacy, 10% a traditional healer. 6% had no treatment. 34% of CDWs had to work through sickness. Work conditions Average work hours—9 h per day One third work 9–10 h per day, 28% work 7–8 h per day and 23% work more than 11 h per day 87% have ≥3 h per day as break time Nearly all work 7 days per week 80% can leave the job if they want to 91.3% are allowed to visit home, 74% are allowed to meet friends 90% sleep at their employers’ houses, 72.8% said that their sleeping place is better than home, 2.6% stated that it is not good as home
2	Alem 2006	Non-labourers were significantly more likely to have a self-reported and confirmed mental disorder than non-labourers.	Behavior and mental problems The prevalence of self-reported mental health problems was nearly two times higher among non-laborers (14%) compared with child-laborers (8.5%), *p* < 0.001; OR * = 1.86, 95% CI ** = 1.24–2.77 Specific disorders were higher in non-laborers vs. laborers: Excessive fear (3.0% vs. 0.9%, *p* = 0.002); Retardation (4.0% vs. 2.0%, *p* = 0.017); Elimination problems (2.8% vs. 1.3%, *p* = 0.027). However, this pattern was not significant when comparing child domestic workers against non-workers (no figures reported). Domestic workers also reported difficulty with getting on with others compared with non-laborers (2.5% vs. 0.3%, *p* = 0.006). 5.5 of children with self-reported problems met the diagnostic criteria for a mental disorder Prevalence of confirmed mental disorder diagnosis was higher in non-laborers (8.8%) versus child laborers (4.9%, *p* = 0.002). The common emotional problems in child laborers were phobias (3.1%), enuresis (1.0%), and separation anxiety (0.4%).
3	Banerjee 2008	42.2% of CDWs had experienced different types of abuse Over 35% of CDWs had varying levels of nutritional status impairment.	Violence Abuse: beating 18.8% (62), rebuke 16.6% (55); mental assault 3.3% (11); sexual abuse 3.4% (12). Health Disease pattern and nutrition: Gastro-intestinal tract infection: 72.1% (238); skin disease: 53.3% (176); anemia: 52.4% (173). Nutritional status: 54.8% (154) had grade I malnutrition, 1.8% had grade II malnutrition (height for age); 45.6% had grade III malnutrition (weight for age); 20.6% had mild malnutrition, 9.3 had moderate malnutrition, 5.7 had severe malnutrition (weight for height)
4	Benvegnu 2005	Prevalence of behavioral problems was high in working children, younger working children, and those in the domestic sector.	Behavioral problems Younger children (10–13 years): Prevalence of behavioral problems was higher in workers (21.4%) compared with non-workers (15%); APR *** = 1.3, CI ** = 0.9–1.9, *p* = 0.228. Older children (14–17 years): Prevalence of behavioral problems was lower in workers 9.5% compared with non-workers (12.8%); APR *** = 0.6, CI ** = 0.4–1.0, *p* = 0.042 Prevalence among younger working children (21.4%) was more than double that of older working children (2.5%); APR *** = 2.7, CI **= 1.4–5.1, *p* = 0.003. Children performing domestic services had more behavioral problems than those who did not work; APR *** = 1.6, CI ** = 1.0–2.7, *p* = 0.052.
5	Budlender and Bosch 2002	None reported sexual harassment at work 8% of CDWs were injured while doing activity The most common complaints of both CDWs and child workers in other sector were tiring work and long working hours	Workplace injury 8% of CDW vs. 4% of children in other sectors were injured during work. No CDWs vs. 2% of children in other work sectors reported illness caused/worsened by activity. Work conditions 16% (8715/53,942) of children engaged in paid domestic work reported long working houses (beyond age-specified work hours) 3% of children (≥10 years) worked 43 h or more per a week The most common complaints of CDWs were tiring work (29%), long hours (17%), and fear of being hurt (13%) 3% of CDW vs. 4% of children in other sectors often did heavy physical work.
6	Degraff 2016	43.2% (1129/2608) of children engaging in risky work were in the domestic sector CDWs (6.57%) vs. those in other forms of risky work (8.2% in street work, 13.85% in construction, 14.87% in hazardous farming) had experienced injury/illness from work	Workplace injury/illness Among CDWs, injuries and illness were more common amongst older children aged 15–17 years (7.56%) vs. younger ones aged 10–14 years (4%) Work conditions CDWs work 41.44 h per week 42.44% had to use machines/chemicals at work, while 35% received training or safety equipment.
7	Fassa 2005	Self-reported musculoskeletal pain in preceding 12 months was higher in CDWs compared to non-workers. Over a quarter of CDWs had neck pain (27.8%) and knee pain (25.6%)	Musculoskeletal problems The prevalence of musculoskeletal pain (at any anatomical site) was higher amongst domestic workers than non-workers: APR *** = 1.17, CI = 1.05–1.31. Prevalence of back pain was higher amongst CDWs compared with non-workers: APR *** = 1.23, CI ** = 1.04–1.45.
8	Gamlin 2015	Prevalence of self-reported health and psychosocial outcomes were reported for individual countries	Violence Among CDWs, 49% Togolese, 35% Indians and no Peruvians were physically punished, 58% Filipino were “just talked to” when they made mistakes In Togo, the number of CDWs who know someone who has been physically/sexually abused is twice that of their non-CDW counterparts; In India, ~25% of CDWs vs. 1.2% of controls know someone who has been abused. Health Prevalence of CDWs reporting good or very good health: India (36%); the Philippines (65%); Togo (46%); Tanzania (80%); Peru (51%); Costa Rica (not reported). Filipino control children had highest psychosocial scores, while Togolese CDWs had the worst. (No summative scores, no *p*-values or confidence intervals reported.). Compared with their non-CDW counterparts, Indian and Togolese CDWs had low psycho-social outcomes while Peruvian, Philippine and Tanzanian CDWs had high levels of psychosocial satisfaction Work condition The majority of CDWs in India, Togo and Tanzania worked 10–12 h per day, six or seven days a week. 91% of CDWs in India, and 72% in Togo reported that they do not have any days off in the week. 49% have no free time at all in their working day.
9	Garnier 2003	62.9% (134/213) of migrant maids had experienced disease during the past three months Advanced sexual maturity and better nutritional status amongst non-migrants compared with migrants.	Health and care during sickness 61.5% (204/332) of study participants reported illness during the preceding three months No significant difference in the prevalence and type of illness between migrants (58.8%, 70/119) and non-migrants (62.9%, 134/213), x^2^—0.538, *p* = 0.463. 60% of migrants vs. 34% of non-migrants worked during sickness, x^2^—0.267, *p* = 0.606 Migrants compared to non-migrants are more likely to pay for health services themselves. Development and nutritional status Adolescents living in more socio-economically advantaged environments had more advanced puberty than those living in less socio-economically advantaged environments (difference in breast development stages: x^2^ = 20.78; *p* = 0.008; difference in menarche occurrence: x^2^ = 11.02; *p* = 0.004). Migrants had a more advanced puberty status than non-migrants: breast development according to Tanner’s stage (*p* = 0.045) and occurrence of menarche (*p* = 0.014). After controlling for the effect of sexual maturation on nutrition and growth, migrants had a higher mid-arm circumference (*p* < 0.001), body mass index (*p* < 0.001), and fat mass index (*p* < 0.0001) and a lower stature (*p* < 0.0001) than non-migrants. Work conditions On average, CDWs worked for 10 h (range: 3–15) per day.
10	Gilbert 2018	Youths who have worked as restaveks before age 18 had a higher rate of childhood violence rather than their counterparts who had never worked as restaveks Former female restaveks had higher reported levels of physical, emotional, and sexual violence compared with their male counterparts.	Violence Physical violence among former restaveks vs. non-restaveks: Females: 76.8% (70.4–83.2) vs. 61.9% (56.5–67.3), *p* < 0.0001; Males: 69.6% (60.2–79.0) vs. 62.6% (57.5–67.6), *p* = 0.1661 Emotional violence among former restaveks vs. non-restaveks: Females: 54.8% (47.2–62.3) vs. 33.4% (29.9–36.9), *p* < 0.0001; Males: 51.4% (41.5–61.4) vs. 25.7% (22.2–29.2), *p* < 0.0001 Sexual violence among former restaveks vs. non-restaveks: Females: 39.9% (33.2–46.5) vs. 26.3% (23.3–29.4), *p* = 0.0001; Males: 32.4% (22.6–42.1) vs. 20.5% (17.4–23.6), *p* = 0.0111 Odds of reported violence Female former restaveks vs. controls: Physical (OR * = 2.04, 95% CI ** = 1.40–2.97); Emotional (OR * = 2.41, 95% CI ** = 1.80–3.23); Sexual (OR * = 1.86, 95% CI ** = 1.34–2.58 Male former restaveks vs. controls: Physical (OR * = 1.37, 95% CI** = 0.88–2.14); Emotional (OR * = 3.06, 95% CI ** = 1.99–4.70); Sexual (OR 1.85, 95% CI ** = 1.12–3.07).
11	Hesketh 2012	30% (153/500) of CDWs in India vs. 1% (2/200) of CDWs in the Philippines were beaten/deprived of food for punishment In both countries, those with poor psychosocial wellbeing were more likely to have fair/poor self-reported health and be beaten/deprived of food in India.	Punishment 51% (254/500) in India and 18% (36/200) in the Philippines had been scolded 4.2% (21/500) in India and 0% in the Philippines had been given a reduced salary 67% of CDWs in India and 36% of CDWs in the Philippines had lower levels of psychosocial wellbeing compared to 25% and 30% in the control groups, respectively Psychosocial wellbeing Mean total psychosocial score ^1^ (%) CDWs vs. controls: India 17.7 vs. 25.5 (*p* = 0.007); the Philippines 28.6 vs. 29.6 (*p* = 0.8). Percentage with the lowest tertile psychosocial score (CDW vs. control): India 67% vs. 25% (*p* < 0.001); the Philippines 36% vs. 30% (*p* = 0.2). Amongst CDWs in the lowest tertile for psychosocial wellbeing, the odds of reporting fair/poor health vs. good/very good health: India, OR * = 1.4, 95% CI ** = 1.1–1.6, *p* < 0.0001; the Philippines, OR * = 1.8, 95% CI ** = 1.4–2.2, *p* < 0.0001; the odds of being punished for mistakes (beating & deprivation of food): India OR * = 3.6, 95% CI ** = 3.2–4, *p* < 0.0001; NA for PH Work conditions Indian CDWs worked very long hours and only 8.6% had a day off each week. Overall, the Filipino CDWs worked shorter hours than the Indian CDWs and were more likely to get a day off.
12	ILO 2006	14% (*n* = 14) had experienced abuse 36% of CDWs had been sick or wounded during employment, with a higher proportion of younger children reporting this (53.3%) than older children (32.9%). 1 CDW had poor self-assessed health status	Violence: 13% (*n* = 13) were frequently reprimanded by their employers, one CDW was teased and flirted with, while 86% did not report abuse When asking if they knew other CDWs being abused, 30 said ‘Yes’: 8 (26.7%) knew those being oppressed/shouted/reprimanded, 1 (3.3%) knew those beaten, another knew those who were flirted with and 15 knew ‘No’ abused CDWs Health and care during sickness Compared to before working as a CDW, 76% of CDWs said that their health had remained unchanged, 17% said it got better, 1% said it got worse and 6% did not answer Among those who reported the same/better health, the proportion who stayed at their employer’s house (98.7%, 76/77) was higher than among those who did not stay in the employers’ houses (73.9%, 17/23). Common illnesses reported included cough/respiratory problems (33%), back pain (25%), and wounds (cuts, burns, etc.) (11%) 19/36 (53%) of CDWs reported being sick/wounded at work but not receiving treatment. Of those 17 CDW who were treated, 7 saw a doctor/nurse, 5 saw a pharmacist, and 5 were treated by employers Work conditions On average, CDWs worked 12.5 h per day, 7 days per week and 11.58 months per year. 94% worked 7 days per week, 6% worked 6 days per week. 86% worked 12 months per year while 9% worked 11 months per year 11% did not have regular free time In their free time, 47% did not go anywhere, 22% visited friends/relatives/went to public places
13	Kifle 2002	62% (62/100) reported “often” or “sometimes” experiencing inflicted physical violence 92% were frequently cursed at, 91% were frequently insulted/scolded, 80% repeatedly criticized/belittled 62% (52/84) of females reported any kind of sexual harassment 16% (16/100) reported being taken for medical treatment and 78% (8/23) were self-funded for health care.	Physical violence Of those who reported physical violence, 38% (32/84) of females vs. 69% (11/16) of males reported ‘often’ experiencing physical violence. No (0) boys vs. 45% girls reported ‘never’ having experienced physical violence 41% (19/46) females vs. 19% (3/16) males reported serious injuries and accidents: body swelling/bleeding, bruising, seriously hurt and could not work for some days due to violence 10% (10/100) had experienced punishment by starvation Emotional violence Because of the physical violence inflicted, over 90% were often depressed (57/62) and fearful (58/62) Over 90% of CDWs had experienced rows, altercation nagging, frequent scolding, and insults 93% (78/84) of females and 87.5% of (14/16) males had been cursed at, causing fear/distress amongst participants of the ‘curse coming true’ in 91% (71/78) of females and 71% (10/14) of males cursed 83% of CDWs reported that employers’ behaviors affected their feelings in some way causing worry, weeping, lack of sleep and fear. 62% felt inferior compared to others at school or in the community Sexual violence 46% of females (39/84) had experienced some form of sexual abuse: 3.6% (*n* = 3) reported attempted rape; 28.6% (*n* = 24) flirting for sexual relations and 26.2% (*n* = 22) reported molestation causing suspicion (63.2%), fear (73.7%) and worthlessness (39.5%), while 38% (32/84) had never been sexually harassed. These sexual harassments were mainly committed by employer’s sons (sexual violence not reported amongst male participants). 42.6% (35/84) of CDWs feel sexually insecure at home Due to sexually incited behaviors, CDWs felt fear (73.7%), suspicion (63.2%), worthlessness (39.5%), apathy (31.6%), anger (23.7%), and depression (18.4%) Care during sickness 70% (16/23) of those who fell ill reported being taken for medical treatment, and 78% (5/23) paid for treatment themselves Work conditions 15% did not have any rest, and many worked (number not provided) at least 11 h per day, seven days a week None were allowed to leave the premises, except to go to school A very small number were allowed to meet outsiders and play with children of neighbors.
14	NIS Cambodia, ILO 2004	Over 70% of the CDWs reported being “advised or warned” or being scolded for some infraction. Abuse in other forms was reported such as being slapped with bare hands, being beaten with objects or abused with harsh/vulgar words.	Violence 78.2% (21,871/27,950) had been scolded, 74% (20,670/27,950) had been advised/warned, and 3.2% (894/27,950) had been abused with harsh/vulgar words 8.9% (2510/27,950) had been slapped/beaten with bare hands, 2.4% (675/27,950) had been beaten with objects No CDWs reported sexual violence Health 26.4% had been slashed by sharp objects followed by 10.7% who had slipped in the bathroom and 6.2% who had experienced electrical shock. However, not all of these injuries had occurred only while working Approximately 20% of CDWs suffered from exhaustion (23%), fear (21.3%), insomnia (20.3%), and tension (12.3%) (The 293 CDWs interviewed were used to extrapolate estimates of CDWs in Phnom Penh) Work conditions The average number of work hours was 4 h per day, six days per week: 70.7% worked 1–5 h per day, 19.1% worked 6–8 h per day, and 10.2% worked 9–13 h per day. 57% worked seven days per week, 29.3 percent worked only one to five days per week 0.7% took no rest during their workday and received no medical care when sick. 79.8% had uninterrupted sleep. 4.9% did not receive enough food to eat. Over 70% can have friends, spend recreation time with friends, chat with friends and have time for social gatherings with friends
15	Phlainoi 2002	20% of CDWs (*n* = 115) were “sometimes” or “often” punished by employers 7% of CDW reported work-related sickness (8% among girls, 3.8% among boys).	Violence Of those who had been punished, 1.7% had been hit compared to 46% who had received a warning, 0.9% who had been given a salary cut, and 47% who had been neglected Care during sickness When CDWs fell sick, 40% did not do anything, 22% self-treated, 19.2% saw doctors by themselves, 11.5% had employers who took them to doctors, and 6.7% were taken care of by employers directly. Work conditions 15.7 worked >14 h, 47.8% of CDW worked 12–14 h per day, and 30.4% worked 8–11 h per day. 58.3% had to work seven days per week and 41.7% had at least one day off per week 87% and 88.9% responded that work was not hard or heavy. CDWs were expected to be available at all times.
16	Save the Children UK 2006	Approximately 70% were physically abused, the majority were emotionally abused, and one-third were sexually abused.	Physical violence 68.3% of participants (denominator unclear and not all questions had a complete response rate) had faced some form of physical abuse. 5.3% had experienced all forms of physical abuse (including beating and burning). 46.6% had experienced abuse that left them with a bodily injury (25.3% experienced cuts or bruises). 16% had experienced all types of abuse, except burning. Emotional violence 86% (441/513) of CDWs had faced some form of emotional abuse, including shouting (20.1%) or cursing (11.1%). Sexual violence: 20.3% of CDWs (denominator unclear) reported forced sexual intercourse, including 5.7% of male CDWs. 32.2% reported molestation (private parts touched). 22.4% had been made to touch abusers’ private parts 19.5% had been made to watch pornography One third had been emotionally abused by their abusers. 41.5% of violence cases were perpetrated by family members of the employing household, 7% by someone outside employing house (e.g., employer’s neighbor, other CDW) Work conditions On average, CDWs worked 15 h per day with less than 2 h of rest The majority worked everyday Most were allowed to visit their families only once every six months.
17	Zainab and Kadir 2016	8.3% (*n* = 32) of CDWs had experienced physical abuse at their workplace during the past six months One-third of CDWs had an abnormal body mass index (BMI) and 90% had stunted growth	Abuse Physical abuse: 13% had experienced more than one type of physical abuse: slapping on face (60%), hitting with hard object (6%), violent push (6%), restriction of the facilities (6%), hair pull (3%), kicking (3%), twisting of any body part (3%). Health BMI: 67.5% normal weight, 17.9% overweight, 8.1% thin. Stunting: 18.7% mildly stunted, 40.3% were moderately stunted, 31.4% severely stunted (according to height for age) Work conditions 95% of CDW worked overtime more than once per week. 18.2% did not have any day off/week, 80% had 1 day off/week

Note: CDW-child domestic worker, * odd ratio, ** confidence interval, *** adjusted prevalence ratio, vs. = versus, ^1^ Psychosocial scores comprise personal security and social integration, personal identity and valuation, sense of personal competence and emotional and somatic expressions of well-being, BMI-body mass index.

## Data Availability

Data are contained within the article and the Appendix B.

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
