# Peer review of "Child Domestic Work, Violence, and Health Outcomes: A Rapid Systematic Review"

_ijerph, 2021, doi:10.3390/ijerph19010427_

Round 1

Reviewer 1 Report

The present article deals with a very interesting, serious and sensitive issue.
I think it is a very complete article.
Some aspects to improve would be the following:
In the introduction it would be convenient to put in which countries it is legal for children to do domestic work and in which countries this happens more. It would also be interesting to know what is the average age of the children who work and in which gender it is more common.
As for the methodology, although it is very well done, it is a bit outdated as it is only research carried out until summer 2019. Would it be possible to add the most current ones?
The discussion is very good.
I recommend the authors to make a dissemination work with this article since it is a very serious issue and to be improved worldwide

Reviewer 2 Report

1.Taking Child Domestic Work, Violence and Health Outcomes as a research topic is of great significance. It is a very important and worthy research direction, which can be used as a valuable reference in related fields.

2.(L101)This research uses a database for data collection, and excludes some data based on the research theme. However, it is not explained here which keywords are used to search for data. Please explain which keywords are used for search. In addition, please explain the time plan of this study.

3.(L112) This research only points out the part of Inclusion and exclusion criteria, without explaining the reasons, please add an explanation to understand the considerations of exclusion

4.(L125) The article proposes details of the process of data extraction are described in another paper. However, based on this article being an independently published paper, the researcher is also requested to explain here what tools and methods are used to complete the research content of this article

5.Affirm the detailed collation of the data in Table 1 by the researcher, but from the perspective of a research paper, it is expected that the researcher can organize and summarize these narrative data or use the method of comparison to show new ideas, so as to show the data to bring research perspectives contribution

6.(L606) This research presents the contents of many related materials, but the research conclusion is not only a comprehensive description of all the materials, but also a constructive summary. Therefore, the author needs to add more constructive research conclusions to support the arguments brought about by the above research materials.
